# Efficacy of Infrared Vein Visualization versus Standard Technique for Peripheral Venous Cannulation in Infant and Toddler Populations: A Randomized Study

**DOI:** 10.3390/children10101652

**Published:** 2023-10-05

**Authors:** Graham Fehr, Marisa Rigali, Gregory Weller, Shannon M. Grap, Melissa Coleman, Uma Parekh, Vernon M. Chinchilli, Priti G. Dalal

**Affiliations:** 1Department of Anesthesiology, Children’s Hospital of King’s Daughters, Norfolk, VA 23507, USA; graham.fehr@chkd.org; 2Department of Anesthesiology, Virginia Commonwealth School of Medicine, Richmond, VA 23219, USA; marisa.rigali@vcuhealth.org; 3Department of Anesthesiology, Penn State Health, Hershey, PA 17033, USA; gweller@pennstatehealth.psu.edu (G.W.); sgrap@pennstatehealth.psu.edu (S.M.G.); mcoleman@pennstatehealth.psu.edu (M.C.); uparekh@pennstatehealth.psu.edu (U.P.); 4Department of Public Health Sciences, Penn State College of Medicine, Hershey, PA 17033, USA; vchinchilli@pennstatehealth.psu.edu

**Keywords:** anesthesia, intravenous access, toddlers, infrared vein visualization

## Abstract

Establishing intravenous (IV) access in younger patient populations via the traditional cannulation technique for procedures requiring anesthesia is often challenging. Infrared (IR) vein visualization is a modality that aids venous cannulation; however, few reports of this technique exist in the infant and toddler population. The primary aim of this study was to compare the efficacy of IR vein visualization to the standard cannulation technique for obtaining peripheral IV access in infant and toddler populations. Following Institutional Review Board (IRB) approval and written informed consent, children were randomly assigned to either a standard cannulation technique group or an IR vein visualization device group for venous cannulation. The primary outcome variable was the success rate of IV cannulation, and the secondary variables were the total number of attempts and the time to successful cannulation. No difference was noted between either group for first-attempt success rate (standard versus IR: 61.25% vs. 54.4%; *p* = 0.4) or time to establish IV cannulation (standard versus IR: median [interquartile range], 40 s [24–120] vs. 53 s [26–106]; *p* = 0.55). The anesthesiologist’s grading of the anticipated difficulty of IV cannulation was a significant predictor of cannulation success (*p* = 0.0016). Our study demonstrated no significant benefit in utilizing the IR vein visualization device in terms of the overall success rate, number of attempts, and time to establish successful IV cannulation when compared to the standard technique. However, in difficult IV access situations, this device proved to be a valuable rescue adjunct.

## 1. Introduction

Children undergoing anesthesia procedures require intravenous (IV) access. Rapidly obtaining IV access is paramount for the administration of anesthetic, analgesic, and emergency medications, in addition to IV fluids and blood products. The success rate of first-attempt peripheral IV cannulation during anesthesia induction in children varies from 69% to 82% depending on the provider’s experience [1]. The use of various assistive devices, such as trans-illumination with infrared (IR) light and ultrasound, to improve IV cannulation success rates has been reported in the literature [2,3]. Most studies on transillumination techniques included patients from a wide age group [2,4,5,6] and failed to report superiority over the traditional cannulation technique [2,4,5]. Due to increased body fat and patterns of distribution, IV placement is challenging in the younger pediatric population [7]. Though age [1,8] is a known risk factor for difficult IV access and a predictive factor included in the difficult intravenous access (DIVA) score [8,9], few studies have focused on the use of this device exclusively in infant and toddler populations [2,8]. The primary objective of our study was to compare the efficacy of an IR vein visualization device with the standard cannulation technique (i.e., no IR device) in children (age < 2 years) presenting for elective magnetic resonance imaging (MRI) procedures under general anesthesia. Our hypothesis was that using an assistive device, such as the IR vein visualization device, would result in an increased success rate and decreased time to IV placement. This information is helpful in high-risk situations, such as the induction of general anesthesia in infants and toddlers, when establishing rapid IV access is more challenging.

## 2. Materials and Methods

This study was conducted at the Penn State Health Children’s Hospital, Penn State College of Medicine, Hershey Pennsylvania. The hospital is an academic tertiary center with 75 beds and performs about 1500 MRI procedures under anesthesia annually. A total of 6 out of 14 staff anesthesiologists from the Pediatric Anesthesia Division participated in this study. In this randomized controlled study, children (<2 years of age) scheduled for elective diagnostic MRI procedures with general anesthesia were randomly allocated to either an IR vein visualization device group or a standard cannulation technique control group. This study was approved by the Institutional Review Board (IRB #6768) at the Penn State Health Milton S Hershey Medical Center and Penn State Health Children’s Hospital and was conducted at this single institution between July 2017 and October 2020. Parental written informed consent was obtained from all subjects participating in the trial. The trial was registered prior to patient enrollment at clinicaltrials.gov (NCT03181542, https://clinicaltrials.gov/ct2/show/NCT03181542, Principal Investigator: Priti G Dalal, Date of registration: 1 June 2017).

The inclusion criteria were children (<2 years of age) undergoing anesthesia for elective MRI, American Society of Anesthesiologists (ASA) physical status I–III, and parents or consenting adults fluent in written and spoken English. The exclusion criteria were age ≥ 2 years, emergency MRI requiring anesthesia, ASA physical status > III, and patients with pre-existing IV access. The primary outcome variable was the success rate of IV cannulation (with a maximum of 3 attempts). The secondary outcome variables were the total number of attempts and the time until successful IV cannulation.

The study subjects were identified by the investigators (PGD, GW, SMG, MC, and UP) on the day prior to the scheduled procedure, and consent was obtained on the day of the procedure. After obtaining formal written parental consent on the day of the procedure, the child was randomized to either the IR group (IR vein visualization device) or the standard group (traditional cannulation technique) based on a computer-generated number by our staff research technician. In all cases, the attending anesthesiologist (>4 years of clinical experience as a staff pediatric anesthesiologist) performed the IV cannulation after inhalational induction of anesthesia. The study participants were blinded to the technique since the study was performed under anesthesia. The care providers were not blinded due to the nature of the study. A staff research technician noted the data on a data sheet. Prior to induction, the anesthesiologist graded the overall anticipated difficulty of IV cannulation on a scale of 0–5 (0 = no difficulty; 5 = extremely challenging). The anesthesia induction was standardized. Following the application of standard ASA monitors, inhalational induction was performed with a 50:50 mixture of nitrous oxide:oxygen and incrementally increased sevoflurane concentration (up to 8%). Once the child was deemed ready for IV cannulation by the attending anesthesiologist, a standard tourniquet was applied at the muscle belly on the corresponding limb where cannulation was contemplated. For those randomized to the standard group, the traditional technique for establishing IV access was used. For those randomized to the IR vein visualization group, the IR vein visualization device AccuVein (AccuVein, Inc., Huntington, NY, USA) was used as an aid in establishing IV access. In all cases, IV access was attempted with a 24-gauge IV catheter (Protectiv^®^ Safety IV catheters, ICU Medical, Inc, San Clemente, CA, USA) by the attending pediatric anesthesiologist. The data collected included age, gender, ASA physical status, site where IV was successfully established, number of attempts, weight, height, skin color, ethnicity, and pre-existing comorbidities.

The time to successful IV cannulation was defined as the time from the first contact of the needle/cannula unit with the skin until successful cannulation was established, as evidenced using a bolus of 5 mL sterile normal saline via the IV cannula. After the third attempt, if cannulation was still unsuccessful, the alternative technique was used as a rescue technique. The time to successful cannula insertion was noted for each attempt with the alternative technique. Assuming a 90% success rate with the IR vein visualization device, a 95% confidence interval (CI), and a margin of error of 5%, a sample size of 139 patients would need to be enrolled in the study. To account for attrition, we enrolled 160 patients in this study.

Once the study commenced, an interim analysis was performed on 79 study patients, which showed no statistical difference in the efficacy of the two techniques. No study-stopping guidelines were applied.

### Statistical Analysis

SigmaPlot^®^ 12.5 (Systat software, Inc., Palo Alto, CA, USA) was used for descriptive statistics, and statistical analysis software (SAS, SAS 9.4 Copyright © SAS Institute Inc., Cary, NC, USA) was used for the remaining statistical analysis. Descriptive statistics were presented in the form of proportions for binary variables, means with standard deviations for normally distributed variables, and medians with interquartile ranges for non-normally distributed variables. Fisher’s exact test was used for comparing proportions in 2 × 2 tables, and the Jonckheere–Terpstra test was applied when comparing proportions in 2 × K ordinal tables. Binary logistic regression was applied when a potential predictor variable was measured on a continuum. A *p*-value < 0.05 was considered statistically significant.

## 3. Results

A total of 160 patients were enrolled in this study. Of these, one patient in the IR group was excluded because of the inability to complete the study due to staffing logistics. The Consolidated Standards of Reporting Trials (CONSORT) flow diagram is as shown in Figure 1. Both groups were comparable in terms of their demographic data as seen in Table 1. Data on the success rates and times to successful cannulation are shown in Table 2. The overall success rate was slightly higher in the standard group vs. IR group, 90% vs. 85%, respectively, but this was not found to be statistically significant (*p* = 0.34). The time to successfully establish IV access with the primary assigned technique was comparable between the two groups (standard versus IR: median [interquartile range], 40 s [24–120] vs. 53 s [26–106]; *p* = 0.55). The first-attempt success rate was higher in the standard group (61.25%) vs. the IR group (54.4%), although this was not statistically significant as shown in Table 3 (*p* = 0.4). However, the anesthesiologist’s grading from 0 to 5 (0 = not challenging to 5 = most challenging) of the anticipated IV access difficulty was significantly predictive of cannulation success as shown in Table 4 (*p* = 0.0016). One patient in the standard group had their MRI procedure cancelled due to failure of both techniques, with numerous unsuccessful attempts at IV cannulation. None of the patients experienced any unintended harm or adverse events due to IV cannulation.

Overall, the success rate of IV cannulation in both groups was not affected by the presence of lighter-colored skin (*p* = 0.59), history of prematurity (*p* = 0.19), and ASA physical status (*p* = 0.24). The odds ratios for the effect of age, weight, and nil per os (NPO) time for clear liquids on the success rate of IV cannulation were 0.999 (95% CI, 0.996–1.001), 0.98 (95% CI, 0.797–1.205), and 0.999 (95% CI, 0.997–1.001), respectively.

## 4. Discussion

In this randomized controlled study, we found no significant difference in the success rate and time to establish IV access using the IR vein visualization device in comparison to the standard technique in infant and toddler populations under general anesthesia. Most studies in the literature have reported no increase in the success rate or time to cannulation with vein visualization devices [3,4,10]. However, most of these studies are in older age groups [2,3,4]. Our study sought to investigate the effect of this device in a younger patient population (<2 years of age). We chose this age group as there are reported difficulties with establishing IV access in this age group [1]. A previous study has demonstrated higher odds of establishing difficult IV access in children less than 3 years of age compared to older children [1]. We chose 3 attempts for the primary technique before cross-over to the rescue technique since most difficult IV access protocols use >2–4 attempts as the cut-off in their definitions for IV access difficulty [11]. We conducted the study in this patient population based on our previous experience and literature reports of difficult IV access in this age group [11]. Our study corroborates the findings of previous reports on the efficacy of the IR vein visualization device [4,12].

Our findings suggest that the anesthesiologist’s assessment is the best predictor of success with IV placement rather than the device or technique used. This is similar to other reported results [1,9]. Interestingly, we also found that skin color or ethnicity did not affect the success rate of IV access. However, there are mixed reports on these factors [13,14]. A previous study has reported significant influence of age and gender on successful IV cannulation in children [14]. However, our study was a much larger study compared to that study, although we had a smaller proportion of non-white population in our study. A similar study performed in anesthetized children did not show the impact of body mass index (BMI), gender, or racial category in anesthetized children [1]. Perhaps this may be attributed to anesthesia-induced vasodilatation, with the ability to better identify venous structures compared to studies conducted in non-anesthetized children [11,15]. While the IR vein visualization device is useful in identifying the location and trajectory of the vein, it lacks the capability to accurately determine the vein depth under the surface of the skin. Illumination may also cause visual discomfort to the provider attempting to identify the vein. Further, due to the increase and distribution of body fat within our study population, locating deeper veins via IR illumination proved to be a limitation of this device. Thus, the device may prove to be a useful teaching tool for learners. A previous report demonstrated the usefulness of this device as a teaching tool for improving success in IV skills among student learners [16]. A higher success rate was achieved with the IR technique versus the standard technique in students in a simulation setting in that study. The reasons why this device may gain popularity as a teaching tool include easier identification of the vein and realistic visualization of the vein [16].

Prolonged fasting prior to anesthesia procedures may lead to dehydration, resulting in a decrease in intravascular volume, making attempted IV cannulation challenging. In this study, we did not find an increased risk of failed IV attempts associated with an increased duration of fasting for clear liquids. This is in keeping with another study that found no association between fasting time for clear fluids and the number of attempts for a successful IV insertion [17]. This is interesting because prolonged fasting has been attributed to hypotension [17], and some studies have suggested difficulties in establishing IV access. These factors were highlighted in recent literature suggesting a change in guidelines to one hour for clear fluids [18]. However, this was not the primary aim of our study. Perhaps other confounding factors such as patient characteristics may have contributed to this finding. Although the anesthesiologists were familiar with this device, their experience with the device was much less than that of the routinely used standard technique. In the future, increased experience with this device may improve both the success rate and the rapidity with which IV access is established. Recently, the ultrasound technique for establishing IV access has achieved popularity [19]. There are several advantages and disadvantages of either technique. The advantages of the ultrasound technique include better anatomical delineation of the vein, direction of the vein, and depth and orientation of the surrounding structures, which may contribute to higher success rates compared to the traditional technique [18]. However, the use of ultrasound requires additional training, experience, and cost. On the other hand, while the IR technique helps delineate superficial veins, it may fail to identify deeper veins, especially in obese children. Although no formal training is necessary, familiarity with the IR technique is important. Additionally, there is difficulty in judging the depth of the vein with the IR technique. Future studies evaluating the efficacy and learning curves of the ultrasound technique versus the IR technique are warranted. It is possible that the ultrasound technique is suitable for deeper veins, whereas the IR technique is suitable for superficial veins.

Our study had several limitations. First, familiarity with the device was not the same as that of the standard cannulation technique. This may have contributed to the slightly reduced success rate. Perhaps more frequent use of the device would correspond to a significantly improved success rate. The providers who used the device in this study had >4 years of experience as attending pediatric anesthesiologists and were comfortable using the device. We did not set a standard number of times the IR device needed to be used before determining provider proficiency. Perhaps a standardized assessment of proficiency using the IR device may have improved the success rate. Second, we did not report the effect of the patient’s BMI on the success rate. Perhaps using BMI in comparison to weight would have given a better predictability of the success rate based on body habitus. Third, our time definition included the total time required to insert the IV catheter using the randomly assigned technique. Where multiple attempts were made, it is possible that there was some variability and time lost in moving the equipment, identifying the vein, and then attempting subsequent IV cannulation. Finally, our study was conducted in a specific study population, i.e., children less than two years of age, so it may not necessarily be generalizable to the older population. There may be a population bias, and these findings may not be extrapolated to other settings, such as the emergency department, operating room department, or another institution. While there are several studies on the use of the IR technique, our study is unique as it focused on infant and toddler populations, which have been shown to have a higher risk of encountering difficult IV access.

In conclusion, our study was a randomized study that compared the efficacy of the IR technique versus the standard technique, specifically in children less than two years of age. The IR technique did not prove to be superior to the standard IV cannulation technique when implemented by experienced anesthesiologists. Anesthesiologists’ rating of anticipated difficulty with IV access proved to be a significant predictor of successful IV placement. While this study did not prove the superiority of the IR vein visualization technique over the standard technique, the device proved to be a valuable rescue technique. Future studies should focus on using this device as a teaching tool for developing IV access skills among clinical providers.

## Figures and Tables

**Figure 1 children-10-01652-f001:**
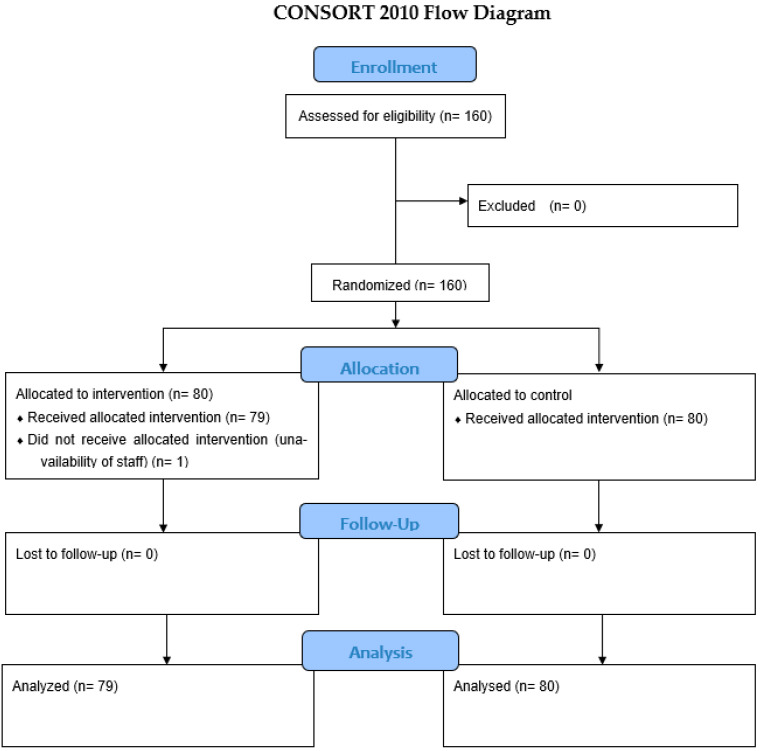
CONSORT flow diagram of participants. CONSORT indicates Consolidated Standards of Reporting Trials.

**Table 1 children-10-01652-t001:** Demographic characteristics.

Characteristics	Standard Group	IR Group	Overall
n (%)	80 (50.3)	79 (49.7) ^a^	159 (100)
Age (months)	12.4 (7.0–17.8)	13.6 (8.1–17.6)	13.5 (7.5–17.6)
Gender, n (%)			
Male	42 (52.5)	43 (54.4)	85 (53.5)
Female	38 (47.5)	36 (45.5)	74 (46.5)
Weight (kg)	9.1 ± 2.2	9.5 ± 2.3	9.3 ± 2.3
Prematurity, n (%)	13 (16.2)	14 (17.7)	26 (16.4)
NPO length (hrs)	4.8 (3.3–9.2)	6.0 (3.5–10.5)	5.3 (3.5–9.4)
Skin Color, n (%)			
Light/Caucasian	57 (71.3)	60 (75.9)	118 (74.2)
Medium	19 (23.7)	15 (19.0)	33 (20.8)
Dark	4 (5.0)	4 (5.1)	8 (5.0)

Data are presented as n (%), mean ± SD, or median (interquartile range). NPO length based on last oral intake of clear fluids. ^a^ One patient was excluded from the IR group as an appropriate research anesthesiologist was not available in time to place the IV. Abbreviations: NPO, nil per os; IR, infrared.

**Table 2 children-10-01652-t002:** Success rate and time to successful IV cannulation outcomes.

	Overall	Standard Group	IR Group
n (%)	159	80 (50.3)	79 (49.7)
Overall success rate, % (n)	87.4 (139)	90.0 (72)	84.8 (67)
First-attempt success rate, % (n)	57.9 (92)	61.3 (49)	54.4 (43)
Subjects requiring rescue technique, n	19	12	8
Rescue technique success rate, % (n)	94.7 (18)	100 (12)	87.5 (7)
Time to successful IV cannulation (sec)			
Primary technique	45 (25.5–115)	40 (24–120)	53 (26–106)
Rescue technique	142 (49.5–357)	142 (82–338)	212 (49.5–534)

Data are presented as n (%), % (n), and median (interquartile range). Reported only first-attempt and overall success rates (included subjects requiring 3 attempts with the assigned technique and then eventual success with the rescue technique). Percentage values listed for rescue technique success rates representative of only this total subset of the study population. Abbreviations: IR, infrared; IV, intravenous; sec, seconds.

**Table 3 children-10-01652-t003:** Total number of IV cannulation attempts.

Number of Attempts	Standard Group	IR Group
n (%)	80 (50.3)	79 (49.7)
1	49 (61.3)	43 (54.4)
2	15 (18.7)	18 (22.8)
3	16 (20.0)	17 (21.5)
>4	0 (0.0)	1 (1.3)

Data are presented as n (%). Number of attempts listed with percentage values of the total for that specific subset of patients. *p* = 0.40, Abbreviations: IR, infrared.

**Table 4 children-10-01652-t004:** Distribution of overall IV cannulation success rate and anticipated IV access difficulty rating by the anesthesiologist on a grade of 0–5 (*p* = 0.0016).

	Success		
Anticipated Difficulty Rating, n (%)	No	Yes	Total
0 (least challenging)	0 (0)	5 (100)	5
1 ^a^	1 (4.4)	22 (95.6)	23
2	1 (2.4)	40 (97.6)	41
3	9 (18.4)	40 (81.6)	49
4	4 (18.2)	18 (81.8)	22
5 (most challenging)	5 (27.8)	13 (72.2)	18
Total	20 (12.7)	138 (87.3)	158

Data are presented as n (%). Subjective grading scale based on perceived difficulty of attempted venous cannulation by the attending anesthesiologist (0 = least challenging; 5 = most challenging). Success rates represent all subjects and are not differentiated into each randomized group (standard or IR). Percentage values represent proportion of the total for the respective difficulty rating. ‘Yes’ represents success with the randomized technique and ‘No’ represents failure with the randomized technique. One subject failed both techniques. Factors such as body habitus and skin color were taken into consideration. *p* = 0.0016. ^a^ One patient did not have a difficulty value assigned by the attending anesthesiologist at the time of their procedure. Abbreviations: IV, intravenous; IR, infrared.

## Data Availability

Minimal data are available upon request due to privacy concerns.

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
