# Peer review of "Efficacy of Infrared Vein Visualization versus Standard Technique for Peripheral Venous Cannulation in Infant and Toddler Populations: A Randomized Study"

_children, 2023, doi:10.3390/children10101652_

Round 1
Reviewer 1 Report
Dear author,
Thank you very much for creating this manuscript on a useful subject.
Your paper is nicely written and easy to understand, congratulation for this.
As a fan of the IR technique for IV cannulation in children, I agree that the experience of the provider is of utmost importance when using these devices.
I have added some suggestions/remarks to improve the quality of your work.
I look forward to seeing your revised version
Best regards
Reviewer
Tables 3 and 4 are difficult to understand. Please modify. I think some information in these tables is missing.
Table 4 needs a proper explanation in the text. In its current form it’s difficult to understand.
Author Response
Tables 3 and 4 are difficult to understand. Please modify. I think some information in these tables is missing.
Able 3 has been modified – the first row was missing, this has been added, page 5, line 161
Table 4 needs a proper explanation in the text. In its current form it’s difficult to understand.
Lease see page 3, line 116
Reviewer 2 Report
Comment for the Authors
I reviewed the manuscript by Fehr et al. entitled “Efficacy of Infrared Vein Visualization Versus Standard Technique for Peripheral Venous Cannulation in the Infant and 3 Toddler Populations: A Randomized Study” submitted to children (Manuscript ID: children-2619283). In this randomized clinical trial at a single institution, the authors mainly investigated the usefulness of the infrared (IR) vein visualization device on the success rate of intravenous (IV) cannulation; total number of attempts; and time to successful cannulation as compared to the standard cannulation technique. They found that all of these measured outcomes were similar between the two techniques. They also observed that the anesthesiologist’s grading of the anticipated difficulty of IV cannulation was a significant predictor of cannulation success. The topic is interesting, the methodology used here seems to be appropriate, and data are presented clearly. Therefore, this reviewer thinks that this manuscript would provide useful information to the readers of this journal. First, the reviewer pays respect for the authors' tremendous effort spent on this manuscript. However, there are several concerns regarding this manuscript, which are listed below. The reviewer believes that the authors can address all of these comments appropriately.
Methods
1
Setting
Please describe settings and locations more details (e.g. tertiary hospital, academic hospital, community hospital, teaching hospital, number of hospital beds and operating room, annual number of general anesthesia, annual number of pediatric anesthesia, and number of anesthesiologists, annual number of sedation for diagnostic MRI procedures, etc) where the data were collected. This information should help readers to depict the context of this study more accurately.
Following description were incomplete or missing according to the CONSORT 2010 checklist. Please recheck:
2
Who generated the random allocation sequence, who enrolled participants, and who assigned participants to interventions? Please provide such information according to the CONSORT guideline.
3
When applicable, please explain any interim analyses and stopping guidelines. If not, please state so.
4
blinding
If done, who was blinded after assignment to interventions (for example, participants, care providers, those assessing outcomes) and how. If not, please state so.
Results
5
Participant flow diagram is missing. Please give the numbers of participants who were randomly assigned, received intended treatment, and were analyzed for the primary outcome. For each group, please explain losses and exclusions after randomization, together with reasons. The authors should provide the flow diagram as Figure 1.
6
Table 1
Generally, P-value is not necessary in this table (patient characteristics). The raw data clearly indicates that there are no clinically significant differences between two groups, and the randomization was working. Whether statistically significant or not is, actually not important. Statistical comparisons are therefore not meaningful: Please delete p-values.
7
Table 3
First row of the table is missing. Please check.
8
Please report the all-important harms or unintended effects in each group
9
Please list the all-rescue technique used.
Discussion
10
Discuss the generalizability (external validity) and implications for practice of the study results. Generally, a single site study limits the generalizability of the findings. In this reviewer's opinion, it might not be possible to extrapolate your findings to other medical institutions. What do you think?
11
What is the strength of this study? limitation section should rewrite as "limitation and strength" section.
12
Please provide the implications for practice, including the intended use and clinical role of the IR device more in details.
Other information
13
Funding
Please specify the sources of funding and other support, and role of funders if any.
Reference
14
Reference numbers are unnecessary duplicated.
15
Data Availability Statement: data unavailable publicly due to privacy
The reviewer thinks that the minimal anonymized data set should be made available, according to the journal’s police. The privacy of the patients surely protected.
Although the several criticisms listed above, this reviewer should however state that it is laudable that this work is derived from huge efforts made by the authors, who are working as the frontline healthcare professionals. The reviewer respects the authors’ time and effort spent on this manuscript, and the authors ‘patience and professionalism in dealing with my comments. This reviewer surely recommend publication if my comments listed above are addressed properly.
Author Response
Please describe settings and locations more details (e.g. tertiary hospital, academic hospital, community hospital, teaching hospital, number of hospital beds and operating room, annual number of general anesthesia, annual number of pediatric anesthesia, and number of anesthesiologists, annual number of sedation for diagnostic MRI procedures, etc) where the data were collected. This information should help readers to depict the context of this study more accurately.
This study was conducted at the PennState Health Children’s Hospital. The hospital is an academic tertiary center with 75 beds and performs about 1500 MRI procedures under anesthesia annually. A total of 6 out of 14 anesthesiologists participated in the study. Please see page 2, line 54-56
Following description were incomplete or missing according to the CONSORT 2010 checklist. Please recheck:
This has been added to figure 1 of the revised manuscript. Please see figure 1 on last page 11.
2 Who generated the random allocation sequence, who enrolled participants, and who assigned participants to interventions? Please provide such information according to the CONSORT guideline.
A research technician generated the random number based on computer generation. This has been corrected in the revised manuscript. One of the 5 anesthesiologists listed obtained the consents and enrolled patient. Page 2, 71-73
3 When applicable, please explain any interim analyses and stopping guidelines. If not, please state so.
Once study commenced, an interim analysis was done on 79 study patients that showed no statistical difference in the efficacy of the two techniques. No study stopping guidelines were applied. Page 3, 97-98
4 blinding
If done, who was blinded after assignment to interventions (for example, participants, care providers, those assessing outcomes) and how. If not, please state so.
The study participants were blinded to the technique since the study was performed under anesthesia. The care providers were not blinded due to the nature of the study. A staff research technician noted the data on a data sheet. The care providers were not blinded due to the nature of the study. Page 2, lines 76-77
Results
5 Participant flow diagram is missing. Please give the numbers of participants who were randomly assigned, received intended treatment, and were analyzed for the primary outcome. For each group, please explain losses and exclusions after randomization, together with reasons. The authors should provide the flow diagram as Figure 1.
This has been provided in the revised manuscript as Figure 1. Page 11
6 Table 1
Generally, P-value is not necessary in this table (patient characteristics). The raw data clearly indicates that there are no clinically significant differences between two groups, and the randomization was working. Whether statistically significant or not is, actually not important. Statistical comparisons are therefore not meaningful: Please delete p-values.
This has been deleted in the revised manuscript.
7 Table 3
First row of the table is missing. Please check. The first row has been added in the revised version, page 10, line 158
8 Please report the all-important harms or unintended effects in each group
9 Please list the all-rescue technique used.
One patient in the standard group had their MRI procedure cancelled due to having failed both techniques with numerous unsuccessful attempts at IV cannulation. None of the patients had any unintended harm or adverse events due to IV cannulation.
This sentence has been added to the manuscript. Page 9, lines 118-119
Discussion
10 Discuss the generalizability (external validity) and implications for practice of the study results. Generally, a single site study limits the generalizability of the findings. In this reviewer's opinion, it might not be possible to extrapolate your findings to other medical institutions. What do you think?
Our study was in a specific study population i.e. children less than two years of age so may not necessarily be generalizable to the older population. There may be population bias and these findings may not be extrapolated to another setting such as emergency or operating room department or at another institution. This has been addressed in the revised manuscript – in the study limitations section. Page 8, lines 261-264
11 What is the strength of this study? limitation section should rewrite as "limitation and strength" section.
While there are several studies on use of IR technique our study is unique as it focused on the infant and toddler population which has been shown to have higher risk of encountering a difficult IV access.
This sentence has been added to the revised manuscript in the limitations paragraph, Page 10, lines 264-265
12 Please provide the implications for practice, including the intended use and clinical role of the IR device more in details. Page 10, lines 241-249
Other information
13 Funding
Please specify the sources of funding and other support, and role of funders if any.
Funding was internal from the Department of Anesthesiology. This has been clarified in the revised manuscript.
Reference
14 Reference numbers are unnecessary duplicated.
This has been addressed in the revised manuscript.
15 Data Availability Statement: data unavailable publicly due to privacy
The reviewer thinks that the minimal anonymized data set should be made available, according to the journal’s police. The privacy of the patients surely protected.
Minimal data may be available upon request. This has been addressed in the revised manuscript. Although the several criticisms listed above, this reviewer should however state that it is laudable that this work is derived from huge efforts made by the authors, who are working as the frontline healthcare professionals. The reviewer respects the authors’ time and effort spent on this manuscript, and the authors ‘patience and professionalism in dealing with my comments. This reviewer surely recommend publication if my comments listed above are addressed properly.
The authors wish to thank the reviewer for bringing up these important points and giving us the opportunity to revise the manuscript.
Round 2
Reviewer 2 Report
Thank you for your effort. The revision is satisfactory and this reviewer has no further comments or suggestions.